# Modelling the Kerf Angle, Roughness and Waviness of the Surface of Inconel 718 in an Abrasive Water Jet Cutting Process

**DOI:** 10.3390/ma16155288

**Published:** 2023-07-27

**Authors:** Marcin Płodzień, Łukasz Żyłka, Krzysztof Żak, Szymon Wojciechowski

**Affiliations:** 1Department of Manufacturing Techniques and Automation, The Faculty of Mechanical Engineering and Aeronautics, Rzeszow University of Technology, W. Pola Str. 2, 35-959 Rzeszow, Poland; plodzien@prz.edu.pl; 2Department of Manufacturing and Materials Engineering, Faculty of Mechanical Engineering, Opole University of Technology, Mikolajczyka Str. 5, 42-271 Opole, Poland; k.zak@po.edu.pl; 3Institute of Mechanical Technology, The Faculty of Mechanical Engineering, Poznan University of Technology, Piotrowo 3, 60-965 Poznan, Poland; szymon.wojciechowski@put.poznan.pl

**Keywords:** surface roughness, surface waviness, abrasive water jet cutting, kerf angle, modelling

## Abstract

An experimental study of the abrasive water jet cutting process of Inconel 718 alloy samples with varying values of cutting speed, abrasive flow rate and cutting material height was carried out. Surface roughness and waviness were measured at different cutting depths, and the variation of the kerf angle was studied. It was shown that the depth of cut has the greatest effect on roughness and waviness. The height of the sample has no impact on the roughness and waviness at a particular depth of cut. As the depth of cut increases, in most cases, roughness and waviness increase as well. It has been proven that the cutting speed has a negligible effect on surface roughness, but it has a significant effect on surface waviness. The waviness, on the other hand, depends only slightly on the abrasive flow. It has been proven that the kerf angle does not depend on the abrasive flow. The kerf angle depends mainly on the height of the sample. The models were developed for the parameters of roughness Ra and Rz, waviness Wa and Wz and kerf angle. All models were calculated without separating the surface into smooth and rough cutting regions.

## 1. Introduction

Nickel superalloys have applications in various industries, such as the aerospace, energy and chemical industries [1]. Their common use results mainly from their unique properties, mostly high strength at high temperature resistance (up to 700 °C) as well as corrosion and fatigue resistance [2]. Therefore, nickel alloys are used predominantly for the production of parts that operate at high temperatures. Critical components of aircraft engines and gas turbines are an example of such parts. The unique properties of nickel alloys negatively affect their machining, which is why they are classified as difficult-to-cut materials. Processing of these alloys by cutting, abrasion and eroding proves to be a challenge and requires detailed research and the selection of appropriate tools and cutting parameters. Therefore, it is impossible to apply the available traditional machining process models to describe the cutting process of nickel alloys.

One of the unconventional machining processes is abrasive water jet (AWJ) cutting. This process is increasingly used in industrial practice due to the lack of heat effects on the workpiece material, as well as high machining efficiency and no negative environmental impact [3]. The most important quality indicators of AWJ cutting include surface roughness and waviness and cutting accuracy [4]. The surface quality varies depending on the cutting zone. In many scientific papers, the cut surface is divided into two zones [5]. In the upper zone, called the smooth cutting region, the dominant parameter is the surface roughness and in the lower zone, called the rough cutting region, where the jet curves, the dominant parameter is the surface waviness [6]. The accuracy of the workpiece and the surface after cutting depends on the kerf angle [7]. This is due to the fact that the kerf has a wider entry and its width decreases as the jet cuts into the workpiece [8].

The parameters of the AWJ process have a significant impact on the quality of the workpiece [9]. The most important cutting parameters include cutting speed, abrasive flow rate, type of abrasive, abrasive grain size, water pressure, stand-off distance SOD and height of the workpiece [10]. Depending on the design and control of the AWJ cutting machine, in practice, it is very often impossible to control all parameters of the cutting process [4,11]. However, each parameter has a significant impact on the quality of the workpiece.

The most important quality parameter of the workpiece are the surface roughness and waviness. The study of surface roughness and waviness after AWJ cutting has been the subject of many research works concerning the machining of various materials, including Inconel 718 alloy. In work [12], cutting process of the Kevlar fiber-reinforced polymer was studied. The study revealed that the traverse speed had the greatest influence on the surface roughness, which was modelled separately for the top- and bottom-cutting zone. The study of surface roughness after cutting was also conducted by Akkurt et al. [13]. Tests were conducted for various nonferrous materials for two material thicknesses, 5 and 20 mm. Only selected roughness parameters were tested. Surface waviness was not tested. It was proved that the surface roughness of material of 5 mm thickness is higher than 20 mm thickness for materials based on brass and steel. The surface roughness of the Inconel 625 alloy was tested in work [14]. It has been shown that stand-off distance is the most influencing parameter among the input tested parameters such as pressure, stand-off distance and abrasive flow rate. Inconel 625 alloy was also the subject of research [15]. The influence of process parameters such as the abrasive flow rate, pressure and gap distance on surface roughness and kerf angle was determined. It was found that the water pressure is the dominant factor in the created equation. Results opposite to those in the work [14] were obtained. On the other hand, in the cutting of Inconel 188 alloy, it was shown that the cutting speed does not have a significant influence on the surface roughness but the surface roughness increases with increasing stand-off distance and decreases slightly with increased abrasive flow rate [16]. Various studies have also examined the surface roughness of titanium alloys, which are also classified as hard-to-machine materials. Obtained results were very similar to those obtained for nickel alloys. For example, the work [17] shows that the abrasive flow rate contributed to 29.32% and the stand-off distance to 61.77% in controlling surface roughness. Another study of roughness as a function of depth of cut was conducted but only for an aluminum alloy [18]. The effect of the abrasive mass flow rate and the cutting speed on the roughness parameter Ra was studied. The surface roughness was shown to change slightly with increasing abrasive mass flow rate. No roughness models were developed and only the Ra parameter was studied. Model of surface roughness as a function of process parameters was developed and also in [19]. It was proved that the abrasive flow rate and the cutting speed had the strongest effect on the surface roughness. Similar results were obtained in cutting natural stone [20]. It was proved that increasing the cutting speed causes a corresponding increase in the roughness and waviness of the surface. The cutting tests on stainless steel conducted in [21,22,23] showed that cutting speed directly affects surface roughness. With decreasing cutting speed, the cut surface quality improves visibly.

Workpiece dimensional quality depends on the kerf. The larger the kerf angle, the greater is the machining error, and as a result, the surface of the workpiece after cutting is not parallel to the jet stream and Z axis. A number of papers have investigated the effect of AWJ process parameters on kerf angle. Kumar et al. studied the inclination of the kerf in the AWJ cutting of Inconel 718 [24]. They developed a kerf taper model in which cutting speed had the greatest effect on kerf, and abrasive flow rate had the least effect. On the contrary, paper [25] has shown that water pressure has the greatest influence on the cutting quality of Inconel 718 alloy, and the contribution of this parameter in the developed model exceeded 45%. Inconel 718 was also studied in the article [19]. Model of kerf angle as a function of cutting process parameters like water pressure, abrasive flow rate, cutting speed and abrasive grain size has been developed. It was proved that the cutting speed is the most important factor in the kerf angle model, similar to works [12,25]. Studies on the influence of cutting speed on kerf were also conducted for ceramic tiles to determine the appropriate cutting speed value to achieve the required kerf size [26].

Most scientific research can lead to the conclusion that the stand-off distance, cutting speed and the water pressure have a deciding influence on the surface roughness and kerf angle after abrasive water jet cutting. However, many works lead to contradictory conclusions. Most papers have not focused on the influence of cutting depth on surface roughness and waviness, especially for Inconel 718 alloy. There are no available models of surface roughness and waviness and kerf angle for Inconel 718 alloy, which would take into account not only the process parameters but also the depth of cut and the height of the workpiece without dividing the machined surface into zones.

The purpose of this study was to conduct a comprehensive experimental investigation of the AWJ process of Inconel 718 alloy using variable process parameters such as cutting speed, abrasive mass flow rate, cutting height and depth of cut. Based on the results of the study, empirical models were developed for the roughness parameters Ra and Rz, waviness parameters Wa and Wt and kerf angle as a function of variable process parameters. The developed models allow determination of the surface roughness and waviness and kerf angle at any depth of cut.

## 2. Materials and Methods

The tests were carried out on a KNUTH Hydro-Jet Eco 0615 cutting machine manufactured by KNUTH Werkzeugmaschinen GmbH, Wasbek, Germany equipped with a multiplier pump. The water pressure was 250 MPa. The abrasive was supplied through a belt system. The focusing nozzle had a diameter of 0.3 mm and the cutting nozzle had a length of 80 mm and a diameter of 0.76 mm. The stand-off distance value was equal to 1.5 mm. The abrasive used was Garnet 80. The variable parameters of the cutting process were the cutting speed, abrasive flow rate and the height of the workpiece. The constant and variable cutting parameters are listed in Table 1. A view of the machine workspace is shown in Figure 1. The cutting was carried out in one axis, always in the same direction.

The test samples were made of Inconel 718 alloy. Nine test specimens were made, and each sample had five sections from S1 to S5 with different cut heights *h* (Figure 2). The first section S1 had a height of 2 mm, and each following section was 2 mm higher.

The experimental research was carried out according to a complete two-factor plan with three variables. During the experimental tests, nine parallel cuts were machined with an abrasive water jet at a distance of 5 mm. As a result, nine test specimens denoted from W1 to W9 were cut with variable process parameters according to Table 2. The third variable parameter of the cutting was the height of the workpiece *h*, which resulted from the variable geometry of each sample. Each sample was divided into sections from S1 to S5 with increasing cut height.

After each sample was cut, the measurements of the selected surface topography parameters Ra, Rz, Wa and Wt were conducted. Ra is the arithmetic mean height of profile roughness, Rz describes the maximum height of profile roughness, Wa is the arithmetic mean height of profile waviness and Wt describes total height of profile waviness. For each sample and each section, the surface roughness and waviness parameters were measured, starting from a cutting depth of 0.3 mm with a step of 0.5 mm, until the end of the sample. This way, the results of surface roughness and waviness were obtained as a function of cutting speed, abrasive flow rate, height of the workpiece and actual depth of cut *a_p_*. The *a_p_* depth of cut parameter is the result of the adopted experimental methodology and is the depth at which the roughness and waviness of the surface were measured at the height of the workpiece. Measurement of surface topography was performed with contact method using the MahrSurf GD 120 profilometer by Mahr GmbH, Goettingen, Germany.

The kerf was measured using a Dino Lite AM7915MZT digital microscope. Measurements of the upper and lower kerf sizes were carried out using the dedicated DinoCapture 2.0 software.

## 3. Results and Discussion

### 3.1. Influence of Cutting Height on Surface Topography

Each test sample contained five sections differing in material thickness and the height of the cut *h*. In each section, the roughness and waviness parameters were measured, from a depth of 0.3 mm, with a step of 0.5 mm. The height at which the surface topography parameters were measured corresponded to the actual cutting depth *a_p_*. Section S5 of each sample had cutting areas from the full range of cutting depths *a_p_*, from 0 to 10 mm.

The purpose of the analysis was to investigate the significance of the influence of the sample height on the surface topography obtained at each cutting depth. Accordingly, for each of the nine samples, models were developed for the roughness parameters Ra and Rz and the waviness Wa and Wt as a function of the height of the workpiece *h* and the current cut depth (depth of measurement). Then, an ANOVA analysis of the obtained models was performed to determine the significance of the effect of these two parameters. Table 3 shows the values of the contribution of the workpiece height parameter *h* in each model. When the significance level of the parameter *h* was less than 0.05, the contribution did not occur.

From the data presented in Table 3, it can be seen that the influence of the height of the workpiece *h* on the roughness and waviness of the surface is marginal. In almost half of the models obtained, the parameter *h* is not present at all, which means that it is statistically insignificant. In the remaining cases, the level of contribution of height *h* to the models is only 1–2%. Therefore, it can be concluded that the analysis of section S5 in each test sample is sufficient for the purpose of the surface topography study. Section S5 has a surface with full range of cutting depths, and its analysis is sufficient. Thus, the surface roughness and waviness depend only on the actual depth of cut. The value of roughness and waviness at a given depth of cut does not depend on the total thickness of the material being cut.

### 3.2. Modelling of Surface Roughness Parameters

Since it has been proven that the height of the workpiece does not affect the surface roughness and waviness at a certain depth of cut, the measurement data obtained from the S5 section of each test sample was obtained to develop the experimental models. Measurement of roughness and waviness parameters was carried out over the entire depth of cut with a step of 0.5 mm. As a consequence, results were obtained for the full range of the tested depths of cut *a_p_*.

Mathematical models of the best fit were developed for the two surface roughness parameters Ra and Rz and also the two waviness parameters Wa and Wt. A variance analysis was performed. For each roughness and waviness parameter, an equation was determined, and statistical parameters were determined to evaluate the resulting models. The significance level of α = 0.05 was adopted, so all components of the equations with significance below the accepted level were removed.

First, the surface roughness parameter Ra was analyzed, for which the following equation was obtained:Ra = 4.91 − 0.0497∙*m_a_* + 0.174∙*a_p_* + 0.0123∙*v_f_* + 0.000338∙*m_a_*^2^ + 0.1128∙*a_p_*^2^ − 0.00929∙*m_a_∙a_p_* − 0.000544∙*m_a_∙v_f_* + 0.02077∙*a_p_∙v_f_*(1)
which fits very well with the experimental data, with the coefficient R^2^ = 0.84. A graphical representation of the model is shown in Figure 3. Table 4 shows the results of the variance analysis of the developed model.

Analysis of the Ra parameter model indicates the influence of all three technological parameters on surface roughness. There are linear, two-way and quadratic factors in the model. As can be seen in Table 4, all components of the model are characterized by a high statistical significance of influence, as the values of the *p*-value parameter were close to zero. Only two components of the equation *m_a_*^2^ and *m_a_·v_f_* are characterized by lower significance as their *p*-value parameter reaches 0.01. Nevertheless, considering the *p*-value of the equation components and the total error values, the obtained equation can be considered of a very good fit to the real data. Analysis of the contribution and F-value parameters provides information about the contribution of the components in the equation and their significance of influence on the studied parameter. The analysis of Table 4 shows that the depth of cut *a_p_* has the strongest influence on surface roughness. Its contribution to the equation is more than 50%, and the F-value for the components *a_p_* and *a_p_*^2^ is the largest. The second most significant influence of the cutting parameter is the abrasive flow rate *m_a_*. Its contribution to the model is greater than 10%. On the other hand, the cutting speed *v_f_* proved to have the least significant influence on the surface roughness.

The conclusions of the analysis of Table 4 are confirmed by the graphical representation of the model in Figure 3. In both graphs, it can be seen that the obtained curves have the greatest slope in the direction of variation of the parameter *a_p_*. This means that the surface roughness depends the most on the depth of cut. Furthermore, the dependence of the Ra parameter on the depth *a_p_* is non-linear and varies depending on the other parameters of the cutting process. It can be observed that for the lowest cutting speeds, there is a local minimum of the function of the Ra parameter at a depth of 2 mm. On the other hand, as the cutting speed increases, the relationship becomes monotonic, and the lowest roughness is obtained for the lowest depth of cut and the highest cutting speed. Figure 4a also shows that the higher the cutting speed, the more significant is the influence of the depth of cut on the surface roughness, which is confirmed by the greater angle of slope of the curve in the *a_p_* direction. Also, the abrasive flow rate parameter has a varied influence on surface roughness. Analyzing Figure 4b, it can be observed that for the lowest values of cutting depth, its influence is nonmonotonic. For depths of cut up to 4 mm, the lowest roughness is achieved for an average abrasive flow rate of 120 g/min. On the other hand, for higher values of cutting depth, the lowest roughness occurs for the highest abrasive flow rate. However, the depth of cut has the greatest influence on surface roughness.

Then, the surface roughness parameter Rz was similarly analyzed. A relationship was obtained describing the parameter Rz in a form similar to the equation describing the parameter Ra:Rz = 25.58 − 0.244∙*m_a_* + 0.067∙*a_p_* + 0.154∙*v_f_* + 0.001677∙*m_a_*^2^ + 0.4397∙*a_p_*^2^ − 0.03663∙*m_a_∙a_p_* − 0.003096∙*m_a_∙v_f_* + 0.0939∙*a_p_∙v_f_*(2)

The obtained equation fits the experimental data very well, with the coefficient R^2^ = 0.83. A graphical representation of the model is shown in Figure 4, and Table 5 shows the results of the analysis of variance of the developed model.

Analysis of the *p*-value parameter shows that all components of the equation have a statistically significant influence on the Rz parameter. Moreover, the proportion of components in the model is very similar to the model of the Ra parameter. The depth of cut has the strongest influence on the Rz parameter, and its contribution is almost 50%. The second parameter in terms of contribution to the model is *m_a_*. The influence of the cutting speed is the smallest. Furthermore, the analysis of the graphs in Figure 4 leads to the same conclusions as the analysis of the Ra parameter model. The influence of the cutting speed depends on the depth of the cut. For small cutting depths, the Rz parameter decreases with increasing speed *v_f_*, whereas for larger cutting depths the influence of *v_f_* is negligible. Similarly to the Ra parameter, the influence of abrasive flow rate on the Rz parameter, for small depths of cut, is nonlinear and nonmonotonic. For larger cut depths, increasing the abrasive flow results in lower roughness. However, the influence of depth of cut *a_p_* is the largest, although in many ranges of technological parameters this effect is nonlinear and nonmonotonic.

### 3.3. Modelling of Surface Waviness Parameters

The abrasive water jet is dispersed and curved in the material. This causes not only the deterioration of the surface roughness with increasing depth of cut but also the formation of waviness. Hence, two surface waviness parameters Wa and Wt were examined to determine their variation as a function of the technological parameters of the AWJ cutting process. The following equation was obtained for the waviness parameter Wa:Wa = 8.39 − 0.1877∙*m_a_* − 1.204∙*a_p_* + 0.2145∙*v_f_* + 0.001316∙*m_a_*^2^ + 0.2459∙*a_p_*^2^ − 0.02091∙*m_a_∙a_p_* − 0.002482∙*m_a_∙v_f_* + 0.08263∙*a_p_∙v_f_*(3)

The equation fits the experimental data very well because of the obtained coefficient R^2^ = 0.9. On the other hand, for the waviness parameter Wt, the equation can assume the form presented below:Wt = 42.3 − 0.873∙*m_a_* − 3.97∙*a_p_* + 0.694∙*v_f_* + 0.00557∙*m_a_*^2^ + 0.961∙*a_p_*^2^ − 0.08362∙*m_a_∙a_p_* − 0.00830∙*m_a_∙v_f_* + 0.3235∙*a_p_∙v_f_*(4)

This equation also fits the experimental data very well with the coefficient R^2^ = 0.9. A graphical representation of the Wa model is presented in Figure 5, and the results of the analysis of variance of the generated model are shown in Table 6. In turn, Figure 6 presents graphs of the Wt parameter model, and Table 7 contains the results of the variance analysis. Analysis of the obtained models leads to a conclusion that they are very similar, have the same parameters and differ mainly in the values of the coefficients. This shows that both waviness parameters vary very similarly depending on the technological parameters. This is also proven by the graphs in Figure 5 and Figure 6. Their analysis shows that both Wa and Wt parameters have the same variation as a function of the cutting parameters, *a_p_*, *v_f_* and *m_a_*. Therefore, their common analysis was carried out.

An analysis of the values of the statistical parameters shown in Table 6 and Table 7 proves that in waviness models all parameters show a high statistical significance of influence. For each component of the model, the *p*-value was equal to 0.000. However, in the case of waviness, a different contribution of the model components was obtained compared to the surface roughness models. The depth of cut *a_p_* has the greatest influence on waviness, but its contribution in the model is about 30%, whereas in the case of roughness, its contribution exceeded 50%. As opposed to the models of roughness parameters, in the case of Wa and Wt models, the contribution of cutting speed is very large, almost 30%. This means that the influence of cutting speed on surface waviness is almost as strong as the influence of cutting depth. The contribution and F-value of both parameters are similar. In contrast, the influence of abrasive flow on waviness is the smallest. The confirmation of this analysis is provided by the graphs presented in Figure 5 and Figure 6. It can be seen that for the lowest depths of cut, the waviness does not depend on the cutting speed. The function is almost constant.

As the depth of the cut increases, the cutting speed has greater influence on the waviness. The waviness increases monotonically with increasing speed *v_f_*. On the other hand, for the lowest cutting speed, the minimum of the waviness parameter function can be observed at a depth *a_p_* = 4 mm. It follows that for the lowest cutting speed, the function of waviness parameters has a completely different shape than in the rest of the range. Also, the influence of abrasive flow rate on waviness is not constant and nonmonotonic over the whole range of technological parameters. For the lowest depths of cut, the function has a local minimum for the middle value of the abrasive flow rate. For higher values of depth of cut, it can be observed that the greater the abrasive flow, the lower the surface waviness is.

Images of the surface after AWJ cutting were also analyzed with different abrasive flow rates and cutting speeds. Surfaces with a height of *h* = 10 mm were analyzed to observe changes in surface quality as a function of the cutting depth *a_p_*. Images of the surface after the AWJ cutting are shown in Figure 7.

When analyzing the surfaces after cutting, the main focus has to lie on determining the roughness and waviness parameters because these are the main quality indicators of the cut. Pictures of the surface after cutting confirm the conclusions formulated earlier. It can be seen that as the cutting speed increases, the quality of the surface deteriorates and the irregularities after the abrasive water jet become more and more visible. Furthermore, the higher the cutting speed, the more pronounced is the difference between the upper and lower zones of the cut surface. The surfaces in Figure 7a–c are uniform, and no two zones can be discerned in them. For higher cutting speeds, Figure 7d–i, the differences are more pronounced. In addition, it can be observed that the higher the abrasive flow rate, the better the overall surface quality, with less visible furrows. Furthermore, for the highest cutting speed and lowest abrasive flow rate values, defects were observed on the surface, near the lower edge of the workpiece (Figure 7g,h). This means that for these sets of technological parameters, there is a risk of damage to the Inconel 718 material at depths below about 7 mm.

### 3.4. Modelling of Kerf Angle

Another very important quality parameter of the AWJ cutting process is the kerf (Figure 8). The geometry of the kerf affects the geometric accuracy of the workpiece and the accuracy of the surface after cutting. One of the typical features of AWJ cutting is the formation of a convergent cutting kerf, which depends on the parameters of the cutting process [7,12,26]. The kerf created in the AWJ process is usually described by the angle of convergence *φ.* This angle refers to the taper that occurs on the cut surface due to the nature of the water jet stream (Figure 8). The kerf can be described by the angle *φ*, which can be determined from the relationship:*φ* = arctg (*W_in_* − *W_out_*)/2*h*(5)
where *W_in_* is the width of the gap at the entrance of the jet into the material, and *W_out_* is the width of the gap at the exit of the jet from the material (Figure 8).

In order to determine the kerf angle, the widths of the gap at the entrance of the jet into the material (on the upper surface of the sample) and at the exit of the jet from the material (on the lower surface of the sample) were measured. A view of the measured sample surfaces is shown in Figure 9. Each gap was measured in five locations, and the average value of the width was calculated. Based on the obtained gap widths, the value of the kerf angle was determined using relationship (5).

Figure 10, Figure 11 and Figure 12 show the values of the kerf angle as a function of the cutting speed and the height of the workpiece *h* for different values of the abrasive flow rate *m_a_*. From the presented graphs, it can be seen that the kerf angle *φ* is significantly influenced by the height of the sample *h*. As the height of the material *h* increases, the value of the angle *φ* significantly increases. It can also be seen that the kerf angle values increase with increasing cutting speed, although the increase is no longer so high. Comparing the graphs shown in the following figures, it can be seen that there are no significant differences between them as to the value and trend of the change in the kerf angle. This means that the abrasive flow rate has an insignificant influence on the kerf angle, which has been confirmed in other works as well [19,24].

Then, on the basis of the experimental data, the following functional relationship for the kerf angle was obtained:*φ* = 2.385 − 0.3369·*h* + 0.0591·*v_f_* + 0.01604·*h*^2^ − 0.000372·*v_f_*^2^ − 0.002373·*h·v_f_*(6)

The obtained equation is characterized by a very good fit because of R^2^ = 0.93. When calculating the equation, a significance level of 0.05 was assumed, which means that all components of the equation below this level were removed. As a result, it turned out that the kerf angle depends only on the cutting speed and the height of the sample *h*. The abrasive flow rate *m_a_* statistically does not influence the kerf angle. For a detailed analysis of the obtained model, its graphical interpretation is presented in Figure 13, and the parameters of the analysis of variance are presented in Table 8.

An analysis of Figure 13 and Table 8 indicates that the height of the cut *h* has the greatest influence on the kerf angle. The contribution of the height *h* to the model is almost 80%, and the F-value for the *h* parameter is more than seven times higher than for the other parameters of the equation. The quadratic and two-way factors are of minor importance, as their joint contribution to the model is less than 5%. Despite the very good fit of the model of more than 90%, some components of the model have rather high *p*-value values, which indicates their weaker statistical significance. The cutting speed proves to have a minor influence on the kerf angle, but the influence is statistically significant and noticeable in the graph (Figure 13). The angle of surface slope in the direction of cutting height *h* is significantly larger than in the direction of cutting speed *v_f_*. In addition, it can be seen that as the cutting speed increases, the gap angle *φ* increases less and less. In the case of cutting height, the trend of increasing does not change.

## 4. Conclusions

An experimental study of the AWJ process was conducted. Selected surface roughness and waviness parameters were analyzed as a function of cutting process parameters such as cutting speed, abrasive flow rate, height of cut sample and depth of cut. Experimental models of the parameters of Ra, Rz, Wa and Wt were developed and analyzed. The following conclusions can be drawn from the analysis of the surface topography of Inconel 718:The height of the sample has no impact on the roughness and waviness at a particular depth of cut. Regardless of the height of the cut sample, the surface roughness and waviness at a particular depth of cut did not change. Therefore, the cut height parameter *h* does not appear in the models of Ra, Rz, Wa and Wt parameters.The depth of cut has the greatest influence on surface roughness. Abrasive flow rate has a small influence and cutting speed has only a marginal effect.Surface waviness is equally influenced by depth of cut and cutting speed but abrasive flow has a negligible effect.The influence of cutting process parameters on surface roughness and waviness greatly varies and is non-monotonic in certain parameter ranges.Surface roughness and waviness after cutting varies at the depth of cut smoothly without a clear boundary between smooth and rough region

The kerf angle was studied as well depending on the technological parameters of the cutting process. Analysis shows that the kerf angle value is not affected by the abrasive flow rate parameter. The value of the kerf angle did not change significantly for different values of the abrasive flow rate. This was confirmed by the analysis of variance of the kerf angle model, where the *m_a_* factor was removed due to the low significance level of less than 0.05. It was also shown that the value of the kerf angle depends mostly on the height of the material being cut, and this dependence is monotonic. On the contrary, the influence of cutting speed on kerf angle is minor. Changing the speed from 20 to 60 mm/min resulted in a change in the kerf angle of only about 0.3°.

Further research work should focus on expanding the developed roughness, waviness and kerf angle models with the influence of water pressure and stand-off distance.

## Figures and Tables

**Figure 1 materials-16-05288-f001:**
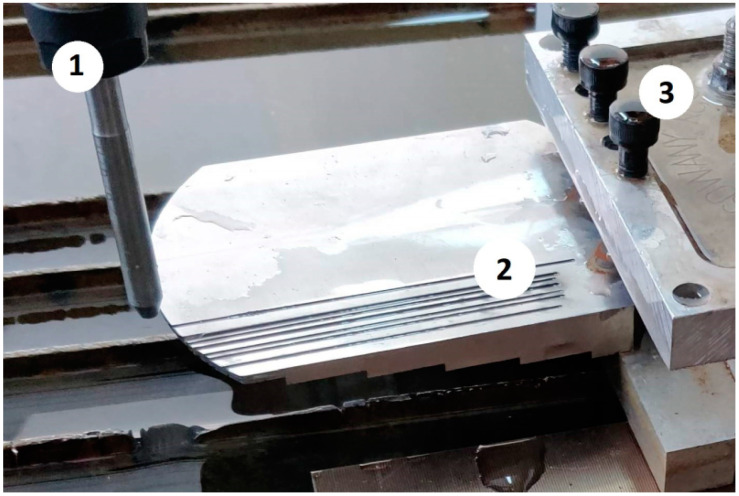
The view of workspace: (**1**) Cutting nozzle; (**2**) Workpiece; (**3**) Clamping unit.

**Figure 2 materials-16-05288-f002:**
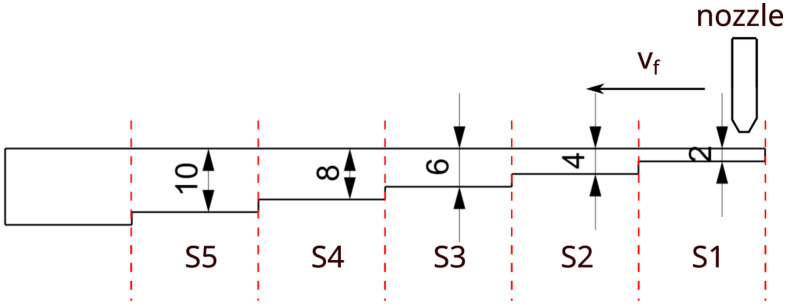
Geometry of the test sample.

**Figure 3 materials-16-05288-f003:**
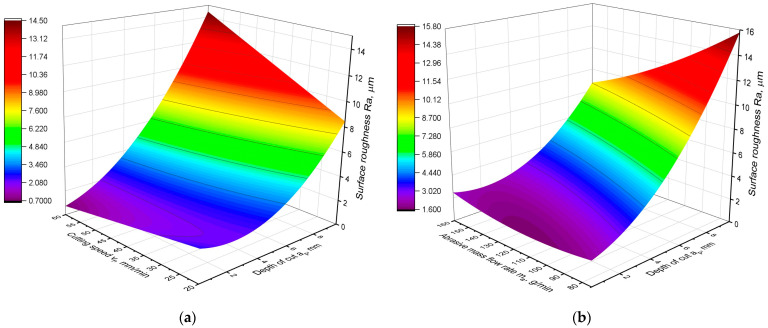
Graphical representation of the Ra parameter model: (**a**) Ra as a function of *a_p_* and *v_f_*; (**b**) Ra as a function of *a_p_* and *m_a_*.

**Figure 4 materials-16-05288-f004:**
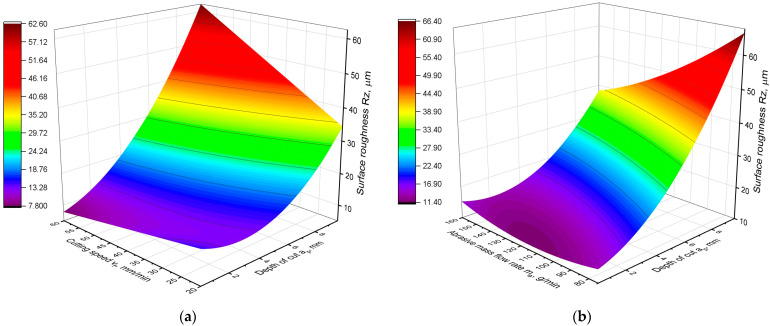
Graphical representation of the Rz parameter model: (**a**) Rz as a function of *a_p_* and *v_f_*; (**b**) Rz as a function of *a_p_* and *m_a_*.

**Figure 5 materials-16-05288-f005:**
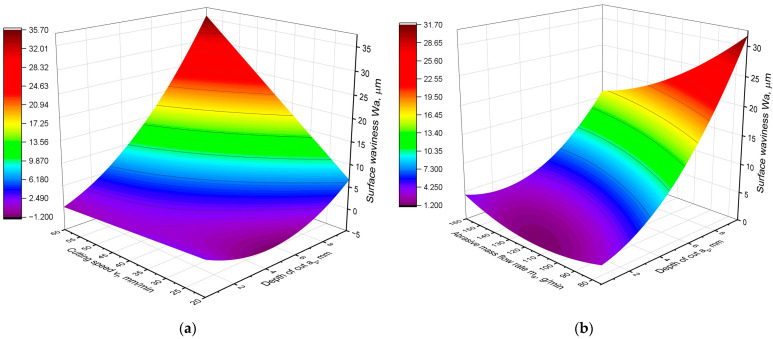
Graphical representation of the Wa parameter model: (**a**) Wa as a function of *a_p_* and *v_f_*; (**b**) Wa as a function of *a_p_* and *m_a_*.

**Figure 6 materials-16-05288-f006:**
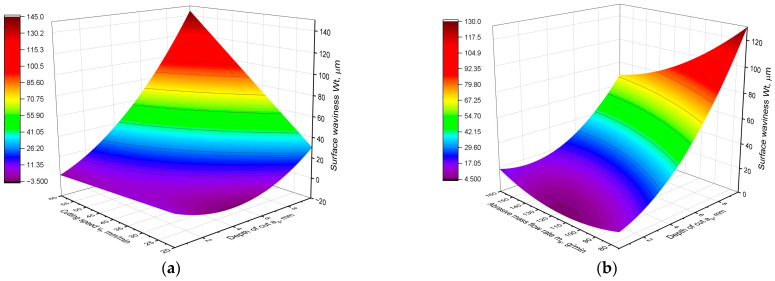
Graphical representation of the Wt parameter model: (**a**) Wt as a function of *a_p_* and *v_f_*; (**b**) Wt as a function of *a_p_* and *m_a_*.

**Figure 7 materials-16-05288-f007:**
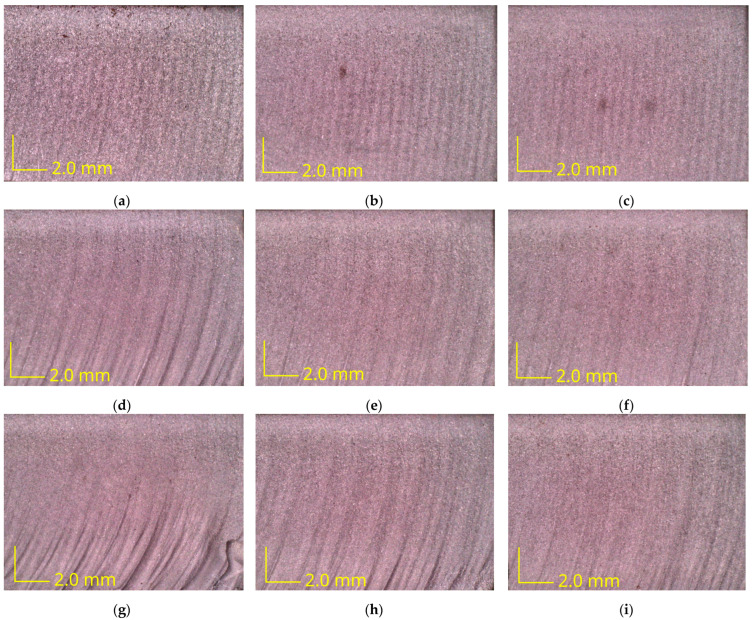
The view of surfaces after AWJ cutting with different process parameters: (**a**) *v_f_* = 20 mm/min and *m_a_* = 75 g/min; (**b**) *v_f_* = 20 mm/min and *m_a_* = 120 g/min; (**c**) *v_f_* = 20 mm/min and *m_a_* = 165 g/min; (**d**) *v_f_* = 40 mm/min and *m_a_* = 75 g/min; (**e**) *v_f_* = 40 mm/min and *m_a_* = 120 g/min; (**f**) *v_f_* = 40 mm/min and *m_a_* = 165; (**g**) *v_f_* = 60 mm/min and *m_a_* = 75 g/min; (**h**) *v_f_* = 60 mm/min and *m_a_* = 120 g/min; (**i**) *v_f_* = 60 mm/min and *m_a_* = 165 g/min.

**Figure 8 materials-16-05288-f008:**
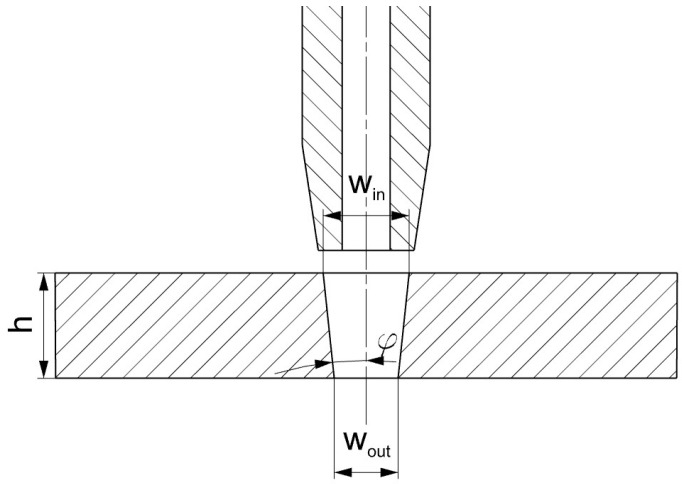
Calculation of the kerf angle *φ*.

**Figure 9 materials-16-05288-f009:**
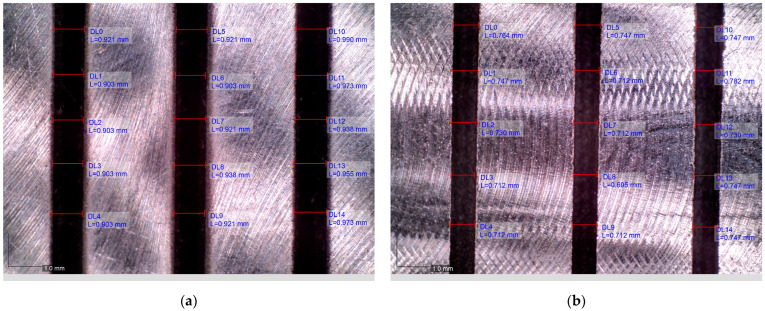
View of the kerf after cutting with *v_f_* = 20 mm/min and *a_p_* = 2 mm: (**a**) *W_in_* in the upper surface; (**b**) *W_out_* in the lower surface.

**Figure 10 materials-16-05288-f010:**
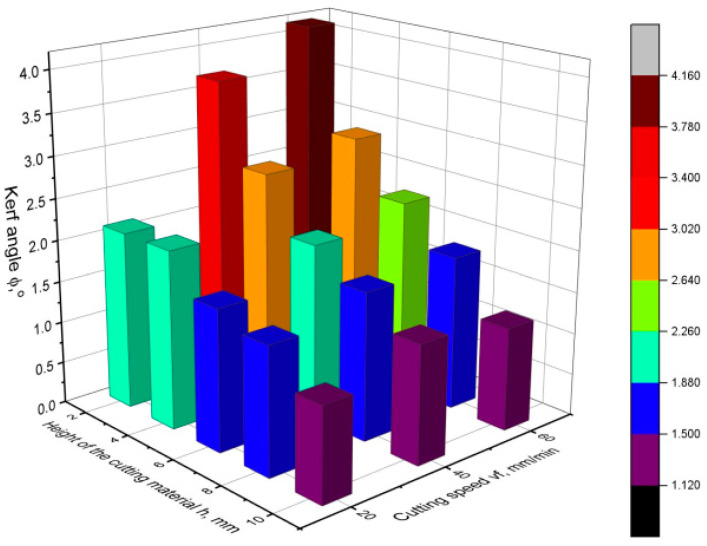
Values of the kerf angle *φ* as a function of cutting speed and cutting height for abrasive flow rate *m_a_* = 75 g/min.

**Figure 11 materials-16-05288-f011:**
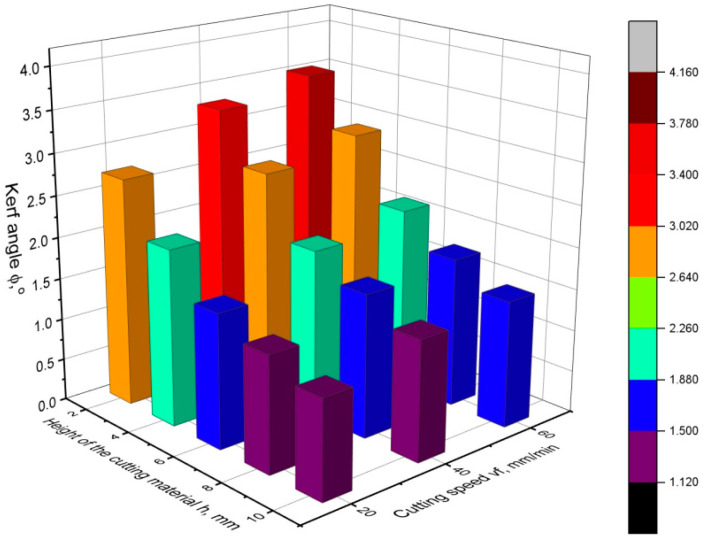
Values of the kerf angle *φ* as a function of cutting speed and cutting height for abrasive flow rate *m_a_* = 120 g/min.

**Figure 12 materials-16-05288-f012:**
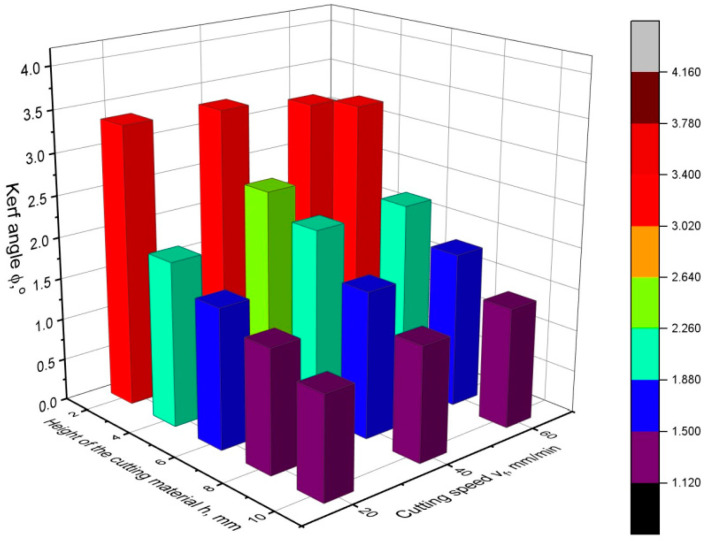
Values of the kerf angle *φ* as a function of cutting speed and cutting height for abrasive flow rate *m_a_* = 165 g/min.

**Figure 13 materials-16-05288-f013:**
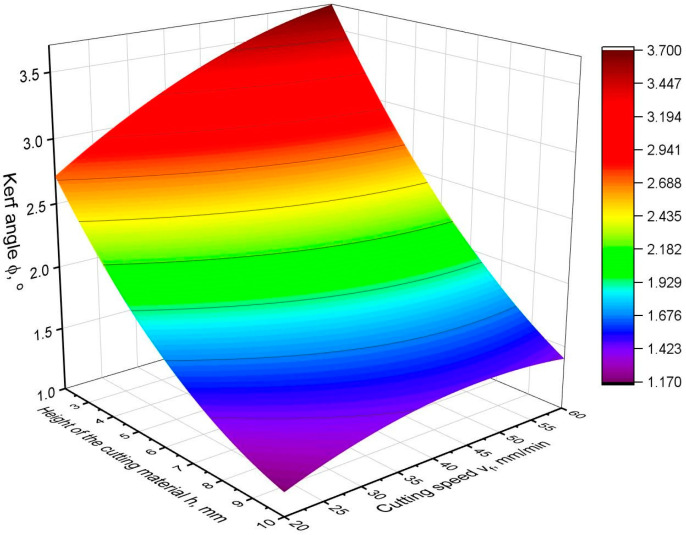
Graphical interpretation of kerf angle *φ* model.

**Table 1 materials-16-05288-t001:** Constant and variable technological parameters.

Constant Technological Parameters	
Abrasive	Garnet 80
Water pressure	250 MPa
Stand-off distance	1.5 mm
Focusing nozzle diameter	0.3 mm
Cutting nozzle diameter	0.76 mm
Cutting nozzle length	80 mm
**Variable Technological Parameters**	
Cutting speed, *v_f_*	20; 40; 60 mm/min
Abrasive flow rate, *m_a_*	75; 120; 165 g/min
Cutting height, *h*	2; 4; 6; 8; 10 mm

**Table 2 materials-16-05288-t002:** Parameters of test samples.

Sample Number	Cutting Speed *v_f_* mm/min	Abrasive Flow Rate *m_a_* g/min
W1	20	75
W2	20	120
W3	20	165
W4	40	75
W5	40	120
W6	40	165
W7	60	75
W8	60	120
W9	60	165

**Table 3 materials-16-05288-t003:** Contribution values of cutting height parameter *h* in models of Ra, Rz, Wa and Wz parameters.

	W1	W2	W3	W4	W5	W6	W7	W8	W9
Ra	0.8%	0.07	0.73	-	-	-	-	-	-
Rz	0.45	-	1.46	-	-	-	-	-	1.7
Wa	1.04	1.62	0.05	0.13	-	-	1.1	-	0.38
Wt	0.08	2.34	0.72	0.4	-	-	0.64	0.39	0.09

**Table 4 materials-16-05288-t004:** Analysis of variance for the Ra surface roughness parameter model.

Term	DF	Seq SS	C (%)	Adj SS	Adj MS	F-Value	*p*-Value
Model	8	2366.78	83.65%	2366.78	295.85	108.71	0.000
Linear	3	1897.73	67.07%	1987.34	662.45	243.41	0.000
*m_a_*	1	294.69	10.42%	334.27	334.27	122.83	0.000
*a_p_*	1	1487.84	52.58%	1557.16	1557.16	572.17	0.000
*v_f_*	1	115.19	4.07%	128.18	128.18	47.10	0.000
Square	2	124.41	4.40%	140.64	70.32	25.84	0.000
*m_a_* ^2^	1	15.90	0.56%	18.66	18.66	6.86	0.010
*a_p_* ^2^	1	108.51	3.83%	122.78	122.78	45.12	0.000
2-way	3	344.64	12.18%	344.64	114.88	42.21	0.000
*m_a_ a_p_*	1	159.97	5.65%	170.06	170.06	62.49	0.000
*m_a_ v_f_*	1	16.84	0.60%	18.88	18.88	6.94	0.009
*a_p_ v_f_*	1	167.83	5.93%	167.83	167.83	61.67	0.000
Error	170	462.65	16.35%	462.65	2.72		
Total	178	2829.44	100.00%				

**Table 5 materials-16-05288-t005:** Analysis of variance for the Rz surface roughness parameter model.

Term	DF	Seq SS	C (%)	Adj SS	Adj MS	F-Value	*p*-Value
Model	8	36,950.8	83.51%	36,950.8	4618.8	107.61	0.000
Linear	3	28,486.8	64.38%	30,020.0	10,006.7	233.13	0.000
*m_a_*	1	4782.0	10.81%	5441.1	5441.1	126.77	0.000
*a_p_*	1	20,859.1	47.14%	22,003.1	22,003.1	512.63	0.000
*v_f_*	1	2845.7	6.43%	3121.0	3121.0	72.71	0.000
Square	2	2022.9	4.57%	2309.6	1154.8	26.90	0.000
*m_a_* ^2^	1	401.1	0.91%	459.6	459.6	10.71	0.001
*a_p_* ^2^	1	1621.8	3.67%	1865.4	1865.4	43.46	0.000
2-way	3	6441.1	14.56%	6441.1	2147.0	50.02	0.000
*m_a_ a_p_*	1	2452.9	5.54%	2641.9	2641.9	61.55	0.000
*m_a_ v_f_*	1	559.8	1.27%	612.6	612.6	14.27	0.000
*a_p_ v_f_*	1	3428.4	7.75%	3428.4	3428.4	79.87	0.000
Error	170	7296.8	16.49%	7296.8	42.9		
Total	178	44,247.6	100.00%				

**Table 6 materials-16-05288-t006:** Analysis of variance for the Wa surface waviness parameter model.

Term	DF	Seq SS	C (%)	Adj SS	Adj MS	F-Value	*p*-Value
Model	8	16,598.0	89.85%	16,598.0	2074.75	188.19	0.000
Linear	3	12,083.1	65.41%	12,797.4	4265.80	386.93	0.000
*m_a_*	1	1184.3	6.41%	1418.7	1418.70	128.68	0.000
*a_p_*	1	5850.4	31.67%	6342.8	6342.80	575.32	0.000
*v_f_*	1	5048.4	27.33%	5301.7	5301.65	480.89	0.000
Square	2	733.9	3.97%	860.0	429.98	39.00	0.000
*m_a_* ^2^	1	250.8	1.36%	283.0	283.03	25.67	0.000
*a_p_* ^2^	1	483.1	2.62%	583.7	583.74	52.95	0.000
2-way	3	3781.0	20.47%	3781.0	1260.35	114.32	0.000
*m_a_ a_p_*	1	768.5	4.16%	861.0	860.96	78.09	0.000
*m_a_ v_f_*	1	356.5	1.93%	393.7	393.66	35.71	0.000
*a_p_ v_f_*	1	2656.1	14.38%	2656.1	2656.08	240.92	0.000
Error	170	1874.2	10.15%	1874.2	11.02		
Total	178	18,472.2	100.00%				

**Table 7 materials-16-05288-t007:** Analysis of variance for the Wt surface waviness parameter model.

Term	DF	Seq SS	C (%)	Adj SS	Adj MS	F-Value	*p*-Value
Model	8	268,094	89.92%	268,094	33,512	189.59	0.000
Linear	3	199,158	66.80%	210,345	70,115	396.66	0.000
*m_a_*	1	16,976	5.69%	20,517	20,517	116.07	0.000
*a_p_*	1	101,766	34.13%	109,773	109,773	621.02	0.000
*v_f_*	1	80,415	26.97%	84,334	84,334	477.11	0.000
Square	2	11,925	4.00%	13,856	6928	39.20	0.000
*m_a_* ^2^	1	4535	1.52%	5064	5064	28.65	0.000
*a_p_* ^2^	1	7390	2.48%	8905	8905	50.38	0.000
2-way	3	57,012	19.12%	57,012	19,004	107.51	0.000
*m_a_ a_p_*	1	12,369	4.15%	13,772	13,772	77.91	0.000
*m_a_ v_f_*	1	3921	1.32%	4406	4406	24.93	0.000
*a_p_ v_f_*	1	40,722	13.66%	40,722	40,722	230.38	0.000
Error	170	30,049	10.08%	30,049	177		
Total	178	298,144	100.00%				

**Table 8 materials-16-05288-t008:** Analysis of variance for the kerf angle *φ* model.

Term	DF	Seq SS	C (%)	Adj SS	Adj MS	F-Value	*p*-Value
Model	5	24.6462	93.31%	24.6462	4.9292	108.81	0.000
Linear	2	23.3655	88.46%	23.3655	11.6828	257.88	0.000
*h*	1	20.6236	78.08%	20.6236	20.6236	455.24	0.000
*v_f_*	1	2.7419	10.38%	2.7419	2.7419	60.53	0.000
Square	2	0.7403	2.80%	0.7403	0.3701	8.17	0.001
*h* ^2^	1	0.5187	1.96%	0.5187	0.5187	11.45	0.002
*v_f_* ^2^	1	0.2216	0.84%	0.2216	0.2216	4.89	0.033
2-way	1	0.5404	2.05%	0.5404	0.5404	11.93	0.001
*h v_f_*	1	0.5404	2.05%	0.5404	0.5404	11.93	0.001
Error	39	1.7668	6.69%	1.7668	0.0453		
Total	44	26.4130	100.00%				

## Data Availability

Data available on request.

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
