# Peer review of "Modelling the Kerf Angle, Roughness and Waviness of the Surface of Inconel 718 in an Abrasive Water Jet Cutting Process"

_materials, 2023, doi:10.3390/ma16155288_

Round 1
Reviewer 1 Report
The reviewed manuscript constructed available models of roughness and waviness and kerf angle for Inconel 718 alloy in an abrasive-water jet cutting process. Authors introduced this work are worth encouraging.
1. Extensive editing of English language is required.
2. The introduction needs a lot of enhancement. The description of the introduction is too wordy and needs to be concise. Also, the purpose of this paper needs to be introduced at the end of the introduction.
3. The experimental data of the regression equation obtained are a little too small, and the coefficient R2 is a little low. So the obtained results are not convincing enough.
Extensive editing of English language is required.
Author Response
Thank you very much for all your comments and suggestions regarding the manuscript. All comments have been considered in the text of the paper. The responses are listed below.
- Extensive editing of English language is required.
In accordance with the reviewer's comment, the text of the manuscript has been checked and linguistically corrected.
- The introduction needs a lot of enhancement. The description of the introduction is too wordy and needs to be concise. Also, the purpose of this paper needs to be introduced at the end of the introduction.
The introduction has been corrected. At the end of the introduction, the purpose of the study is presented.
- The experimental data of the regression equation obtained are a little too small, and the coefficient R2 is a little low. So the obtained results are not convincing enough
This paper presents the real results of tests and measurements of surfaces after the AWJ process. Based on the results, empirical models for the roughness, waviness and kerf angle parameters were developed. In the case of waviness and kerf angle, a model fit of R2>0.9 was obtained. Only in the case of roughness parameters the level of fit was slightly lower but the coefficient R2>0.8. Our research experience shows that in the case of roughness parameters it is very difficult to obtain a model fit level greater than R2=0.9. This is due to the fact that the surface roughness is influenced by a great many factors of the cutting process, especially the abrasive material and the abrasive-water jet. In addition, in the AWJ process the surface roughness varies over the height of the sample depending on the cutting zone, rough and smooth . Models available in the literature have a higher level of fit R2>0.9 because they are roughness models of one cutting zone, usually smooth. In other cases, high levels of fit have been obtained using Artificial Neural Network [26]. In the manuscript, roughness and waviness parameters were modelled for the entire range of cut surfaces, without dividing them into zones. Hence, a lower level of fit was obtained for the roughness parameters.
Reviewer 2 Report
In this study, kerf angle, roughness, and waviness of Inconel 718 are modeled and validated with experimental findings. There is a thorough review of the previous works in the literature, and the novelty of this work is clearly demonstrated by comparison with the previous works. The simulation results are in agreement with the experimental ones. There are a few minor issues that need to be addressed before publication.
1. There is a lot of qualitative information in the abstract, but some significant quantitative results are needed.
2. Due to the large number of parameters, a nomenclature defining each parameter would be helpful to the reader.
3. The letters a, z, and t are notational elements used in Ra, Rz, Wa, and Wt. What do they stand for?
4. For the reader's convenience, the kerf angle should be shown on the figure.
5. The phrase "in the function of" should be replaced by "as a function of".
Author Response
Thank you very much for all your comments and suggestions regarding the manuscript. All comments have been considered in the text of the paper. The responses are listed below.
- There is a lot of qualitative information in the abstract, but some significant quantitative results are needed.
The abstract has been revised and significant information regarding the test results has been added. Results in the form of quantified values have not been added due to the fact that the article concerns the modelling of roughness, waviness and kerf angle parameters. For modelling, the conclusions of the research are mainly in descriptive form.
- Due to the large number of parameters, a nomenclature defining each parameter would be helpful to the reader.
The designations of variable technological parameters in Table 1 have been added. In addition, an explanation of the W1-W9 designations in the manuscript text on page 4 has been added. According to editorial guidelines all parameters were defined the first time they appear in the manuscript text.
- The letters a, z, and t are notational elements used in Ra, Rz, Wa, and Wt. What do they stand for?
The names of the surface roughness and waviness parameters investigated have been added to the text of the manuscript. We have included this information in the “Materials and Methods” chapter. The names of the roughness and waviness parameters are given according to EN ISO 21920-2:2022.
- For the reader's convenience, the kerf angle should be shown on the figure.
The kerf angle is shown in Fig. 8. This figure is in the chapter on kerf angle research. In order to improve the readability of the manuscript, the kerf angle has been described in more detail in section 3.3.
- The phrase "in the function of" should be replaced by "as a function of"
Appropriate changes have been applied to the manuscript.
Reviewer 3 Report
Within this study, the abrasive water jet cutting process of Inconel 718 alloy specimens with varying values of cutting speed, abrasive flow rate, and cutting material height was analized. To establish the effectiveness of the water jet cutting process the surface roughness and waviness at different cutting depths were measured. During the study, 9 parallel cuts were made with an abrasive water jet at a distance of 5 mm. The angle of the kerf was also studied depending on the technological parameters of the cutting process. Analysis shows that the kerf angle value is not affected by the abrasive flow rate parameter and the cutting speed has an insignificant effect on surface roughness. The conclusion of the study is that the depth of cut has the greatest effect on roughness and waviness and also that the cutting speed has a significant effect on surface waviness.
Finally, the reviewer considers that the work might be published in the Materials journal considering the introduction provided enough background information, the research objectives were clearly stated, and the results were presented in a manner that satisfied the standards for a scientific study.
Author Response
Thank you very much for your positive review of the article. We have made changes to the manuscript that take into account the comments of all reviewers.
Round 2
Reviewer 1 Report
Accept in present form
Reviewer 2 Report
As a result of the authors' responses, the manuscript is suitable for publication.